# Disruptive Autoencoders: Leveraging Low-level features for 3D Medical Image Pre-training

**Jeya Maria Jose Valanarasu**[1]                          JMJOSE@STANFORD.EDU
[1] *Stanford University*

**Yucheng Tang**[2]                          YUCHENGT@NVIDIA.COM
[2] *NVIDIA*

**Dong Yang**[2]                          DONGY@NVIDIA.COM
**Ziyue Xu**[2]                          ZIYUEX@NVIDIA.COM
**Can Zhao**[2]                          CANZ@NVIDIA.COM
**Wenqi Li**[2]                          WENQIL@NVIDIA.COM
**Vishal M. Patel**[3]                          VPATEL36@JHU.EDU
[3] *Johns Hopkins University*

**Bennett Landman**[4]                          BENNETT.LANDMAN@VANDERBILT.EDU
[4] *Vanderbilt University*

**Yufan He**[*2]                          YUFANH@NVIDIA.COM
**Vishwesh Nath**[*2]                          VNATH@NVIDIA.COM

**Editors:** Accepted for publication at MIDL 2024

## Abstract

Harnessing the power of pre-training on large-scale datasets like ImageNet forms a fundamental building block for the progress of representation learning-driven solutions in computer vision. Medical images are inherently different from natural images as they are acquired in the form of many modalities (CT, MR, PET, Ultrasound etc.) and contain granulated information like tissue, lesion, organs etc. These characteristics of medical images require special attention towards learning features representative of local context. In this work, we focus on designing an effective pre-training framework for 3D radiology images. First, we propose a new masking strategy called local masking where the masking is performed across channel embeddings instead of tokens to improve the learning of local feature representations. We combine this with classical low-level perturbations like adding noise and downsampling to further enable low-level representation learning. To this end, we introduce **Disruptive Autoencoders**, a pre-training framework that attempts to reconstruct the original image from disruptions created by a combination of local masking and low-level perturbations. We curate a large-scale dataset to enable pre-training of 3D medical radiology images (MRI and CT). The proposed pre-training framework is tested across multiple downstream tasks and achieves state-of-the-art performance. Notably, our proposed method tops the public test leaderboard of BTCV multi-organ segmentation challenge. Our code can be found here.

**Keywords:** Self-Supervised Learning, Auto-encoders, Segmentation.

## 1. Introduction

---

[*] Contributed equally

Inception of transformers (Vaswani et al., 2017; Dosovitskiy et al., 2020) has led to a significant shift from convolution neural network (ConvNet) based methods (He et al., 2016; Ronneberger et al., 2015; Krizhevsky et al., 2017) to transformer-based methods (Dosovitskiy et al., 2020; Liu et al., 2021; Valanarasu et al., 2021; Valanarasu and Patel, 2022; Chen et al., 2021) for many computer vision applications. However, the fact that pre-training plays an irreplaceable role in model development has not changed in the past decade (Radford et al., 2021). Model weight initialization is an important step in training deep neural networks (Kumar, 2017) as good starting weights are necessary for efficient training towards a particular task.

Pre-training for natural computer vision tasks is usually not constricted by the availability of data as natural images are abundant and there is no scarcity and less restrictions in obtaining them. Unfortunately, the

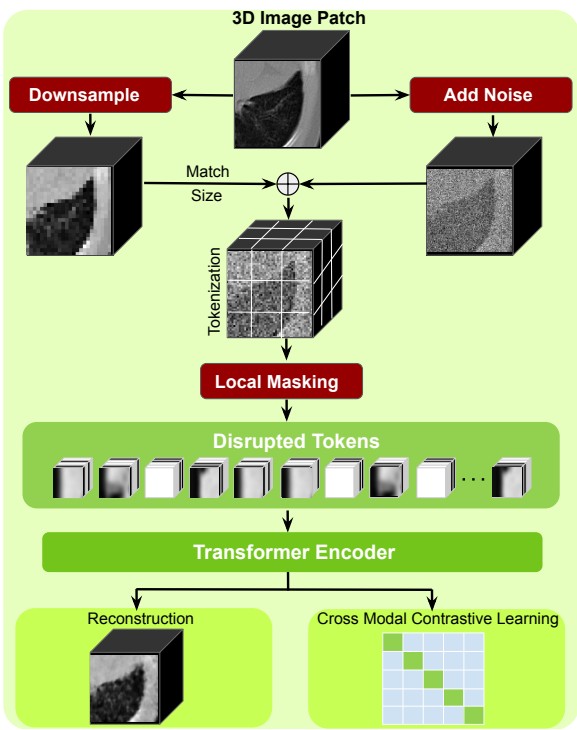

Figure 1: Disruptive Autoencoders.

same does not translate to medical images as they are scarce (acquisition cost is high) and also difficult to obtain (requires specialized hardware). There is complexity involved to release them publicly due to heavy privacy regulations (Saliba et al., 2012).

Recently, masked image modelling methods (He et al., 2022) have gained significant traction as an efficient self-supervised pre-training framework. They are used to develop robust pre-trained models that can generalize well to the downstream tasks. Masked Auto-Encoders MAEs learn a feature representation by trying to reconstruct the original image while masking out randomly selected tokens in the input space. MAEs are designed specifically for transformers as masked tokens help reduce the computation. However, improvements are needed in medical imaging domain: while trying to adopt vanilla MAEs for medical images, we observed that although MAEs do lead to a performance boost for further finetuning, the reconstructions were poor and most of the anatomical structures were missing after reconstruction. This has also been observed in some recent works (Hatamizadeh et al., 2022; Zhou et al., 2022). Unlike natural images, most of the vital information in medical images are in the fine details (e.g. small lesions, finer boundaries of organs, tiny structures of bones that need to be delineated etc).

In this work, we focus on designing an effective pipeline for pre-training on 3D medical volumes. First, we design a new pre-training strategy that is better than MAEs at extracting low-level details. We introduce local masking where we do not mask at the token dimension but at the channel embedding dimension. Unlike MAEs, certain amount of masking is done to all tokens as only the channel embeddings are perturbed (visualized in Fig. 1), helping the network reconstruct sharp details and learn better local context. We also explore

using various low-level vision tasks like denoising and super-resolution for pre-training. We observe that these tasks help extract better low-level features and result in sharper reconstructions (as seen in Fig. 2 it can be seen that MAEs cannot reconstruct the bones and other fine structures while the low-level techniques do) . In summary, we introduce Disruptive Autoencoders (DAE) where we first create a combination of these perturbations (local masking, downsampling, and adding noise) to disrupt the tokens (visualized in Fig. 1). Then, an autoencoder is trained to reconstruct the original medical volume from these disrupted tokens. DAEs result in sharper reconstructions, and a better performance on downstream tasks. We also devise a cross modal contrastive loss for our framework in such a way it can discriminate between the features extracted from different modalities.

In summary, the following are the major contributions of this work:

- We propose Local Masking, a new masking strategy which helps learn useful features to reconstruct local details like small anatomical and functional structures.
- We introduce **Disruptive Autoencoders**, which aim to reconstruct the original volume from tokens disrupted from a combination of low-level perturbations such as local masking, downsampling, and adding noise.
- We curate a public pre-training dataset for CT and MRI radiology images with over 10,000 3D volumes and conduct extensive experiments on multiple segmentation datasets and show state-of-the-art performance. The pipeline achieves best performance on a public multi-organ segmentation challenge leaderboard.

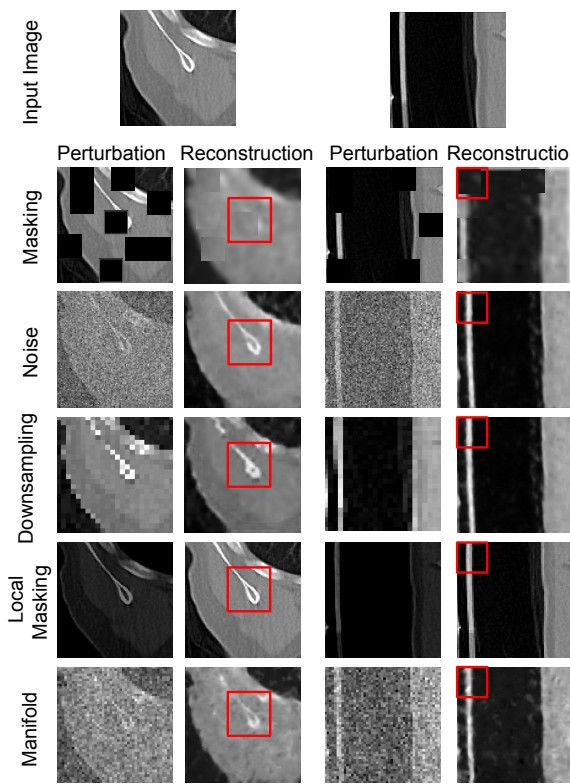

Figure 2: Comparison of reconstructions.

## 2. Disruptive Autoencoders

We propose DAE, a pre-training strategy that focuses on learning strong low-level representative features of the medical image to reconstruct local context. Here, we first take a cubic patch from a 3D medical volume, perturb and tokenize it to get disrupted tokens in 3D. The disrupted tokens are then passed through a transformer encoder to learn a feature representation. The latent features are passed through a decoder and are learned to reconstruct the original image back. The tokens are disrupted using a combination of different low-level perturbations: i) Local Masking ii) Adding Noise and iii) Downsampling. Local Masking is

a novel masking strategy proposed by this work. Denoising and super-resolution (recovering downsampled images) are classic low-level tasks, we are one of the first to explore them as pre-training tasks for medical imaging motivated by the tasks ability to affect low-level finer features of the images. In the next sections, we discuss these in greater detail.

**Local Masking**  The 3D input images are of dimension $(H, W, Ch)$ where $H$, $W$, and $Ch$ denote the height, width, and number of channels in the image respectively. After tokenization, the tokens are of the dimensions $(N, C)$ where $N$ represents the number of tokens and $C$ denotes the embedding dimension. Masked image modelling methods like MAE and SimMIM (Simple Mask Image Modelling) (Xie et al., 2022b) follow a token masking approach where some tokens $X$ out of $N$ are set to zero and the network tries to reconstruct the original image back. The percentage of tokens masked here is a hyperparameter. Token masking approach done in MAEs can be considered a global masking approach as the entire token chosen to be masked is set to zero. To be more specific, the entire $C$ dimension of the chosen $X$ tokens that are to be masked are set to zero.

The entire $C$ dimension, when set to zero, disrupts the image globally, thereby directing the network to learn global context with the objective to reconstruct the original image. This globally disruptive process of setting $C$ does not always help in obtaining a good reconstruction for medical images as most of the information in medical images are not global but in the finer local details. Not being able to learn

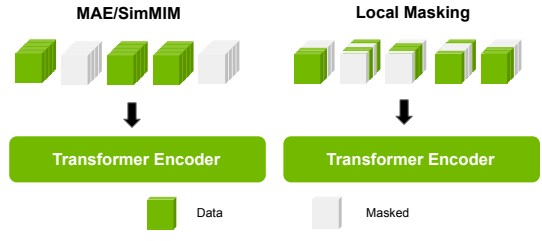

Figure 3: Local Masking

features representative of the local details like anatomy also affects the fine-tuning performance of MAEs. To this end, instead of masking $X$ tokens out of $N$, we propose to mask $X$ channels out of $C$ channel embedding dimension. This ensures that there is some information preserved for each token so that local details do not get completely destroyed. The perturbation is done locally as we set certain embeddings of each token as zero. We call this approach local masking as the masking is done to local details. Local masking has been visualized in Fig. 3. The masking ratio $r$ here is a hyper-parameter which defines the percentage of $C$ embeddings being masked.

**Other Low-level Perturbations**  While local masking helps us extract features representative of local context, we further try to employ other tasks like denoising and super-resolution for pre-training which would help learn more low-level information. Noise and low resolution are commonly found issues in realistic clinical medical acquisition pipelines and hence having them in the pre-training pipeline is meaningful (Zhao et al., 2019).

**Adding Noise:** To obtain a good denoised image, a model must be able to restore all local details of the image like edges and corners. To enable denoising as a pre-training task, we first add noise to the original input and try to restore the original image from the noisy input using the network. Given that the most common additive Gaussian noise, we define the perturbed input $\hat{x}$ as follows:

$$\hat{x} = x + N(\mu, \sigma), \tag{1}$$

where $x$ is the original input and $N$ is the normal distribution. For each sample, we randomly sample from a normal distribution to get the noise. This way, there is no specific pattern for the network to easily restore the input. The mean and variance are hyper-parameters which can be used to control the noise level injected.

**Downsampling:** Super resolution helps generate high-resolution scans from otherwise low-resolution images. To use super-resolution as a pre-training strategy, we first downsample the input image $x$ to get the LR image $\hat{x}$. We formulate this downsampling process as follows:

$$\hat{x} = D(x, \epsilon), \tag{2}$$

where $D$ is the downsampling function and $\epsilon$ is the downsampling ratio. We use linear interpolation for downsampling. We follow a pre-upsampling super-resolution setup where we upsample the LR to same spatial dimension before converting it to HR. The model is trained to recover the HR image $x$ from the LR input $\hat{x}$. Super-resolution is a low-level vision task as we need a features representing fine details to get a HR estimate from blurry LR inputs. Thus, to produce a good HR image, the network needs to learn features that provide rich information to recover all the fine details of the image.

In DAEs, we use a combination of all the above perturbations. We first add noise and downsample the image, then add the two perturbations. Then, the resultant 3D image is tokenized with a local masking strategy. We name these tokens as disrupted tokens and pass it to the transformer encoder. These features are then passed through a decoder for reconstruction and for cross-modal contrastive learning.

**Network and Training:** We use Swin-UNETR (Tang et al., 2022) as our backbone architecture for all the experiments. For pre-training we use a combination of $\mathcal{L}_1$ reconstruction loss and the cross modal contrastive learning loss $\mathcal{L}_{CMCL}$.

First, we explain how to obtain $\mathcal{L}_{CMCL}$. Inspired from (Radford et al., 2021) we train the network to predict which of the $B \times B$ possible pairings across a batch $B$. The goal is to maximize the cosine similarity of the embeddings of the true pairs in the batch while minimizing the cosine similarity of the embeddings of the incorrect pairings. We optimize a symmetric cross entropy loss over these similarity scores.

$$z_{sim} = z_i * z_i^T * exp(t), \tag{3}$$

where $z_i$ is the mini-batch feature vector, $z_{sim}$ is the similarity matrix and $i$ is the mini-batch index. $t$ here is the temperature parameter and is set to 0.07. Note that the total number of data points is $N$ and each mini-batch has $B$ data points which means the mini-batch index $i$ goes from 1 to $N/B$. Now, we apply a simple binary cross entropy loss on the similarity matrix $z_{sim}$ to perform contrastive learning.

We define CMCL loss as follows:

$$\mathcal{L}_{CMCL} = \alpha * CE(z_{sim}, z_{label}), \tag{4}$$

where $CE$ represents the binary cross entropy loss, $\alpha$ represents the scale and $z_{label}$ denotes the labels. CE loss is applied across each axis in the $z_{sim}$ matrix. $z_{label}$ is the label matrix that is created based on the positive pairs and negative pairs as per the meta-data.

The total pre-training loss can be defined as:

$$\mathcal{L}_{pretrain} = \mathcal{L}_1 + \mathcal{L}_{CMCL}. \tag{5}$$

The $\alpha$ in CMCL is set to 0.05 to match the range of both losses. We pre-train Swin-UNETR on a curated set of medical volumes without any labels then use those weights as starting

weights for our downstream fine-tuning experiments. For finetuning experiments, we use a Dice loss to train the model.

## 3. Experiments

### 3.1. Datasets

**Pre-training Dataset:** We various public radiology CT and MRI datasets BraTS21 (Bakas et al., 2018), LUNA16 (Setio et al., 2015), TCIA Covid19 (Desai et al., 2020), HN-SCC (Grossberg et al., 2018), TCIA Colon (Johnson et al., 2008), and LiDC (Armato III et al., 2011) to construct our pre-training dataset. The corresponding number of 3D volumes for brain, chest, abdomen and head/neck volumes are $1,310 \times 4$ (4 modalities), 2,018, 1,520 and 1,223, respectively. The number of brain MRI volumes of each modalities T1, T2, T1ce and FLAIR is 1,310. The total data cohort contains 10001 MR and CT scans of various body region of interests (ROI) such as head, neck, chest, abdomen, brain and pelvis. The details about the pre-training dataset can be found in Table 1.

**Finetuning Dataset - i) BTCV:** Beyond the Cranial Vault (BTCV) abdomen challenge dataset (Landman et al., 2015) consists of abdominal CT scans of 30 subjects. The annotations contain 13 organs which are annotated by interpreters under supervision of radiologists at Vanderbilt University Medical Center. Each CT scan is acquired with contrast enhancement phase at portal venous consists of 80 to 225 slices

| Dataset | Region of Interest | #Total Samples | Train/Validation |
|---|---|---|---|
| BraTS21 - T1 | Brain | 1251 | 1188/63 |
| BraTS21 - T2 | Brain | 1251 | 1188/63 |
| BraTS21 - T1ce | Brain | 1251 | 1188/63 |
| BraTS21 - FLAIR | Brain | 1251 | 1188/63 |
| LUNA16 | Chest | 888 | 844/44 |
| TCIA Covid19 | Chest | 761 | 723/38 |
| HNSCC | Head/Neck | 1287 | 1223/64 |
| TCIA Colon | Abdomen/pelvis | 1599 | 1520/79 |
| LiDC | Chest | 475 | 451/24 |

Table 1: Pre-training Datasets.

with $512 \times 512$ pixels and slice thickness ranging from 1 to 6 $mm$. The multi-organ segmentation problem is formulated as a 13-class segmentation task.

ii) **FeTA**: Fetal Tissue Annotations dataset (FeTA) (Payette et al., 2021) consists of publicly available database of 120 manually segmented pathological and neurotypical fetal MRI T2-weighted brain volumes. These volumes are across a range of gestational ages (20 to 33 weeks) and are segmented into 7 different tissue categories (external cerebrospinal fluid, grey matter, white matter, ventricles, cerebellum, deep grey matter, brainstem/spinal cord). The training images were acquired at two different sites with 1.5T and 3T MR scanners. The data were provided with histogram based matching and zero-padded for $256 \times 256 \times 256$ voxels. Data of both sites was also sampled at 0.5 $mm$ isotropic spacing as per challenge design. The dataset was split into 5-folds of 80/20 for training and validation.

### 3.2. Implementation Details

Our deep learning models were implemented in PyTorch (Paszke et al., 2019) and MONAI. For pre-training experiments, we used a batch-size of 2 per GPU. The volumes were randomly cropped into $96 \times 96 \times 96$ cubes while pre-training. We used an initial learning rate of $4e^{-4}$, momentum of 0.9 and decay of $1e^{-5}$ for 20K iterations. We trained the model using an AdamW (Loshchilov and Hutter, 2017) optimizer with a warm-up cosine scheduler of 500 iterations. We use hyper-parameters $r = 60\%$, $\sigma = 0.1$, $\epsilon = 4$ when training the DAE.

For BTCV fine-tuning experiments, a five-fold cross validation strategy is used to train the models. The models were trained for 600 epochs with a learning rate of $4e^{-4}$ and the batch-size was set to 4 per GPU. Multiple augmentations such as gaussian noise, contrast scaling, zoom and random flipping across the axis were utilized. We select the best model in each fold and ensemble their outputs for final segmentation predictions. For FeTA, the intensities were normalized to a scale of 0 to 1. The learning rate was set to $4e^{-4}$ and batch size was set to 4 per GPU. All models were trained for 600 epochs, which was determined by convergence for the full dataset. Augmentations like random flipping on all 3 axes, Gaussian noise etc. were utilized during the training process. The final layer of the network is also changed from the pre-training configuration to accommodate the fine-tuning task at hand. All pre-training and fine-tuning models are trained using NVIDIA DGX-1 V100 servers with 8 and 4 GPUs, respectively.

### 3.3. Results

We compare our proposed method with previous self-supervised methods like contrastive coding (Chen et al., 2020; Tang et al., 2022), rotation prediction (Taleb et al., 2020; Tang et al., 2022), and masked image modelling methods (He et al., 2022; Xie et al., 2022b). We use Swin-UNETR as our network backbone for all these experiments. We note that MAE and SimMIM are very similar to each other, the only difference being MAEs discard masked tokens while Sim-MIM includes them. So, we just utilize a masked

| Method | HD | MSD | Dice |
|---|---|---|---|
| ResDSN | 24.55 | 1.814 | 0.813 |
| 3D FCN | 38.59 | 4.601 | 0.792 |
| RandomPatch | 18.98 | 1.423 | 0.856 |
| nnUNET | 18.39 | 1.335 | 0.888 |
| UNETR | 39.05 | 1.275 | 0.891 |
| Swin-UNETR | 20.53 | 0.810 | 0.918 |
| DAE (Ours) | **16.82** | **0.654** | **0.921** |

Table 2: Leaderboard [1]results on BTCV. HD: Hausdorff Distance, MSD: Mean Surface Distance.

image modelling configuration but with Swin-UNETR as the backbone and call this configuration MAE in all upcoming discussions. For BTCV, we directly validated our predictions in the public leaderboard and so that we can compare our method with all previous backbone methods. For the leaderboard submissions, we submit to the free competition (no specific registration process required). We train all our models with 80 subjects (20% as validation set), and evaluates on the 20 images test set with spacing resolution of $1 \times 1 \times 1$mm. Within the 80 images, 30 scans are from the public challenge data and 50 extra CT scans annotated by radiologists are used to boost the training performance. We perform 4 rounds of five-fold cross validation experiments and ensemble models to obtain the final prediction. The ensemble process are effective to exclude outliers. In addition, test time augmentation, boundary smoothing and connected component analysis are used for post-processing the labels. Note that this pipe-line for BTCV leaderboard submission is similar to the previous approaches like (Tang et al., 2022) for fair comparison.

These results are tabulated in Table 2. We note that our proposed method performs the best and outperforms all the previous baselines. Specifically, we note that we outperform Swin-UNETR (Tang et al., 2022) which also uses SSL pre-training consisting of 3 different SSL pretext tasks. In particular, we obtain a significant improvement in terms of Hausdorff Distance (HD) and Mean Surface Distance (MSD) compared to previous methods. We also conduct a paired t-test between our BTCV test Dice scores and Swin-UNETR's results.

We obtain a two-tailed p-value of 0.0318 which shows our improvement is statistically significant ($p \leq 0.05$). We also note we obtain better results than even recent methods like UniMiSS (Xie et al., 2022a) which works on the same dataset. Table 3 shows the results of DAE compared with MAE and from scratch on FeTA dataset. In Table 4, we show the comparison of DAE with MAE as well as previous SSL techniques on the five-fold BTCV cross validation dataset. It can be seen that DAE is better than the previous methods.

**Ablation Study:** In Table 5, we conduct an ablation study on DAE. We compare with each of the disruptions separately and then compare against a combination of them. These experiments are conducted on the five-fold BTCV cross validation dataset. For this task, we observe that all three methods perform better than scratch while local masking performs the best out of the three. The combination of the disruptions obtains better performance.

| Data | Method | Fold1 | Fold2 | Fold3 | Fold4 | Fold5 | Avg |
|------|--------|-------|-------|-------|-------|-------|-----|
| 100 % | Scratch | 0.840 | 0.848 | 0.826 | 0.838 | 0.844 | 0.839 |
| | MAE | 0.836 | 0.845 | 0.826 | 0.837 | 0.846 | 0.838 |
| | DAE | 0.843 | 0.850 | 0.826 | 0.841 | 0.850 | **0.842** |
| 50 % | Scratch | 0.654 | 0.693 | 0.650 | 0.645 | 0.667 | 0.662 |
| | MAE | 0.660 | 0.694 | 0.658 | 0.640 | 0.669 | 0.664 |
| | DAE | 0.680 | 0.713 | 0.663 | 0.641 | 0.670 | **0.673** |
| 20 % | Scratch | 0.610 | 0.620 | 0.619 | 0.598 | 0.644 | 0.618 |
| | MAE | 0.615 | 0.640 | 0.610 | 0.616 | 0.642 | 0.625 |
| | DAE | 0.636 | 0.671 | 0.648 | 0.633 | 0.669 | **0.651** |

Table 3: Dice score on 5-fold cross validation on FeTA dataset.

We note that this trend totally depends on the downstream task but with a combination of these perturbations, the pre-trained weights are always better at performance than random initialization.

**Impact of CMCL:** To understand the impact of CMCL, we conduct an experiment on DAE with and without CMCL loss. This experiment is conducted on single fold of the BTCV dataset. It can be observed in Table 6 that CMCL provides a benefit in terms of fine-tuning performance.

| Pre-training Method | Dice ($\uparrow$) |
|---------------------|-------------------|
| Scratch | 0.8343 |
| Contrastive Coding | 0.8367 |
| Rotation Prediction | 0.8356 |
| Swin-UNETR | 0.8472 |
| MAE | 0.8448 |
| DAE (Ours) | **0.8512** |

Table 4: Comparison.

| Pre-training Disruption Strategy | Dice ($\uparrow$) |
|----------------------------------|-------------------|
| Scratch | 0.8343 |
| Noise | 0.8485 |
| Downsampling | 0.8489 |
| MAE | 0.8448 |
| Local Masking | 0.8501 |
| DAE | **0.8512** |

Table 5: Ablation Study.

| Pre-training Strategy | Dice ($\uparrow$) |
|-----------------------|-------------------|
| Scratch | 0.789 |
| DAE w/o CMCL | 0.841 |
| DAE with CMCL | **0.849** |

Table 6: Impact of CMCL.

## 4. Conclusion

In this work, we proposed a new pre-training framework for 3D medical images called Disruptive Autoencoders, where tokens are disrupted using a combination of perturbations: local masking, additive noise, and downsampling. In particular, local masking is a new masking strategy where the masking is performed across channel embeddings instead of tokens to improve the learning of local feature representations. DAE as a pre-training framework performs better than other pre-training strategies across multiple segmentation datasets. Notably, using our proposed method we also achieve the best performance in a public multi-organ segmentation challenge leaderboard and state-of-the-art results on the other public datasets.

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

## Appendix A. Related Works

**Self-supervised Pre-training**: Self-supervised pre-training approaches can be broadly categorized into two types: 1) generative and 2) discriminative. Generative methods focus on mapping the input to a latent space to learn a representation and then decode it back to a new space.

Masked Autoencoders (MAE) (He et al., 2022) propose a way to mask out some tokens and make the network reconstruct the original image back thus helping the model learn useful representative features. It uses an asymmetric encoder-decoder design by having a small transformer decoder to reduce the computation burden. Beit (Bao et al., 2021) proposed a masked image modeling task to pretrain vision transformer while using two views of the input: image patches as well as a discrete visual token. SimMIM (Xie et al., 2022b) simultaneously performs representation encoding and pretext prediction, due to which the decoder design can be changed to be as simple as a single dense layer. Masked feature prediction (Wei et al., 2022) proposes a technique where instead of the original image, manual features like Histogram of Gradients (HOG) are extracted to learn the representation. Latent contextual regressors and alignment constraints have been proposed to map the predicted representations of masked patches to the targets. Masked Pre-training has not been applied only for images but also for point-clouds (Yu et al., 2022), videos (Girdhar et al., 2022), and multi-spectral images (Cong et al., 2022).

Discriminative pre-training methods try to design a discriminative loss to differentiate between the features extracted for different inputs. Typically ground truth in the form of

annotations or labels is not used for pre-training, pretext tasks like solving jigsaw puzzles (Noroozi and Favaro, 2016) or predicting rotation (Gidaris et al., 2018) are used to extract meaningful information. It is also worthy to note CLIP (Radford et al., 2021) uses a contrastive loss for multi-modal data to learn robust visual representations from image and text pairs. Contrastive methods have been shown to be useful in many other multi-modal contexts (Zhang et al., 2022).

**Pre-training for Medical Images:** Model Genesis (Zhou et al., 2019) proposes a unified self-supervised learning framework using the recurrent anatomy in medical images. Azizi et al. (Azizi et al., 2021) perform a stage-wise pre-training on natural images followed by task specific medical images. A Multi-Instance Contrastive Learning based approach is proposed to perform self-supervised pre-training. Several other methods (Kalapos and Gyires-Tóth, 2022) also follow similar contrastive strategies for specific medical imaging tasks. In (Tang et al., 2022), a self-supervised framework using a combination of contrastive coding, rotation prediction, and inpainting was proposed for pretraining on CT volumes. MAE-based pre-training methods have been quickly adopted for self-supervised pre-training on medical images (Zhou et al., 2022; Chen et al., 2022). These works show that masked image modelling methods provides better performance than previous contrastive methods. Unlike these works, we propose a new pre-training setup which efficiently pre-trains on multiple modalities contrastively while also learning all the low-level anatomical details using an autoencoder to have a better representative power. Dira (Haghighi et al., 2022) proposed a SSL method combining discriminative, restorative, and adversarial learning for 2D medical image pre-training. (Hosseinzadeh Taher et al., 2023) proposed an anatomy aware approach to pre-train foundation models for 2D medical images. Lvm-Med (MH Nguyen et al., 2024) and (Nguyen et al., 2023) look into methods combining 2D and 3D effectively for medical image pre-training while Lvm-Med also trained encoders on around 1.3M images of various modalities.

## Appendix B. What do MAEs lack for medical images?

MAEs have shown impressive results for vision based image pre-training. To this end, we first adapted MAEs to operate on 3D volumes for pre-training medical volumes. However, we observed that the reconstruction quality was low as the reconstructions lose the finer anatomical details. Such observations are also seen in other works that try to use vanilla MAEs for medical image pre-training as depicted in Fig. 3. of(Hatamizadeh et al., 2022) & Fig. 2. of (Zhou et al., 2022). Although using these pre-trained weights do improve downstream tasks, we argue that there is significant potential for further improvement in learning better representations as compared to MAEs for medical images. Coarse reconstructions might be sufficient to understand a high-level semantic understanding which is useful for classifying natural images. For tasks like segmentation of medical images, we postulate that coarse features result in poor reconstructions and are not sufficient to enable efficient fine-tuning. MAEs lack in the aspect of learning features that reflect a deeper understanding of the medical image as the tokens are masked globally and no special attention is given to learn the local details. This is a complete version of a proof sketched in the main text.

## Appendix C. Qualitative Results

We also visualize some sample qualitative results of the BTCV dataset in Fig. 4. It can be observed that DAEs result in better segmentation predictions specifically performing better at segmenting small anatomy when compared to MAEs and other baselines.

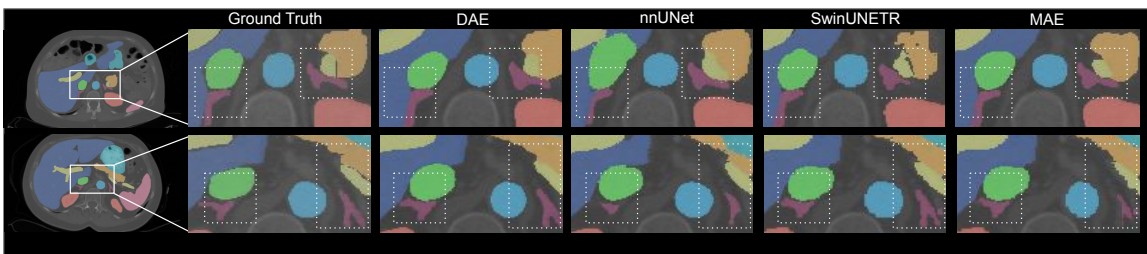

Figure 4: Qualitative visualizations of the proposed DAE and baseline methods for BTCV dataset on two randomly chosen subjects.

## Appendix D. Empirical Analysis to prove low-level features matter

Since a major premise of this work is to improve pre-training pipeline by extracting better low-level features, we conduct a simple experiment to show that low-level features are the most important for fine-tuning. We use CKA (Kornblith et al., 2019) as the feature similarity metric. CKA is used to represent the correlation between any two feature vectors in the latent space. For this experiment, we use MAE and DAE pretrained weights and finetune it on a single fold of BTCV. Now, we feed forward the test images to both the pre-trained model and the fine-tuned model. The CKA calculated between the features extracted from pre-trained model and fine-tuned model across different stages is reported in Table 7. This value gives us an estimate of how much features changed from the initial pre-trained weights to the final fine-tuned model. It can be observed that the deeper layers have the least CKA meaning most of the high-level features have changed after fine-tuning. On the other hand, the early layers have a higher CKA meaning more low-level feature representations were retained from the pre-trained weights to the fine-tuned model. As high-level features undergo a relatively heavier change compared to low-level features after fine-tuning, therefore it is important to focus on learning stronger low-level features during pre-training.

| Stage | 1 | 2 | 3 | 4 | 5 |
|-------|-------|-------|-------|-------|-------|
| MAE | 0.068 | 0.013 | 0.012 | 0.006 | 0.001 |
| DAE | 0.091 | 0.012 | 0.011 | 0.003 | 0.001 |

Table 7: CKA values across different levels of the network calculated between pre-trained and fine-tuned model.

| ID | name | entity | status | team | user+ID | Dice | Mean+Surface+Distance | Hausdorff+Distance |
|---|---|---|---|---|---|---|---|---|
| 9729198 | DAE | syn45147167 | SCORED | lowLevel | @sallyY | 0.9213 | 0.65477 | 16.8232 |
| 9717137 | 3D_SSL_pretrain_SwinTransform... | syn26383386 | SCORED | Anomous | @anonymous_2 | 0.90818 | 1.0868 | 38.4813 |
| 9716420 | UNETR_free | syn26248331 | SCORED | dlmed_nv | @anonymous_ghost | 0.90153 | 1.21 | 38.6018 |
| 9718419 | chen_unetr | syn26842237 | SCORED | | @Lieberk | 0.89047 | 1.2756 | 39.0508 |
| 9712273 | UNETR_ensemble | syn25871870 | SCORED | annoy_ghost | @anonymous_ghost | 0.88512 | 1.2711 | 39.0834 |
| 9706836 | nnunet-dys3 | syn22314225 | SCORED | seu | @JAHe | 0.88345 | 1.132 | 15.56 |
| 9729123 | S_Unet_en | syn44879321 | SCORED | S_UNet | @S_UNet | 0.88013 | 1.2931 | 16.1273 |
| 9727998 | S_Unet | syn43208964 | SCORED | S_UNet | @S_UNet | 0.8796 | 1.408 | 17.6132 |
| 9687073 | CCUhailong | syn20542945 | SCORED | | @longle1319898 | 0.87854 | 1.275 | 17.6623 |
| 9702180 | 3dunet-dys-4 | syn21780923 | SCORED | seu | @JAHe | 0.86351 | 1.8751 | 17.7214 |

Figure 5: BTCV Leaderboard. It can be observed that our method performs the best when compared to all the previous methods.

## Appendix E. Limitations and Future Work

There is scope for improvement in picking the best combination of $\sigma, \epsilon, r$ for the pre-trained weights. A grid search on these parameters would be time-consuming but would lead to better pre-trained weights. Moreover, it can be noted that it takes a lot of time and compute to conduct pre-training experiments with large-scale datasets on dense medical volumes. There is also scope for improvement in increasing the size of the pre-training dataset to obtain more generalizable weights. For future work, we plan on explanding to more modalities like US, XRs, etc.

## Appendix F. BTCV Leaderboard Submission Details

For the leaderboard submissions, we submit to the free competition (no specific registration process required). We train all our models with 80 subjects (20% as validation set), and evaluates on the 20 images test set with spacing resolution of $1 \times 1 \times 1$mm. Within the 80 images, 30 scans are from the public challenge data and 50 extra CT scans annotated by radiologists are used to boost the training performance. We perform 4 rounds of five-fold cross validation experiments and ensemble models to obtain the final prediction. The ensemble process are effective to exclude outliers. In addition, test time augmentation, boundary smoothing and connected component analysis are used for post-processing the labels.

The public leaderboard as of the day of supplementary deadline has been visualized in Fig 5. As there are 100+ submissions, we only display the top performing method in the figure.

## Appendix G. Further Ablation Studies

In the main paper, we conducted ablation studies to estimate the impact of Cross-Modal Contrastive Learning (CMCL) and also the effects of individual perturbations introduced via Disruptive Autoencoders (DAE). Here, we conduct more elaborate ablation studies to

study theimpact of hyperparameters $\sigma$ (denoising), $\epsilon$ (downsampling), and $r$ (masking ratio of local masking) on the downstream performance. All the experiments are conducted on a single fold of Beyond the Cranial Vault Abdominal Segmentation (BTCV) dataset.

### G.1. Impact of Noise level

In Table 8, we study the impact of noise level during pre-training on the fine-tuning performance. The parameter $\sigma$ is the variance of the gaussian noise. Note that in this experiment, we do not include other perturbations of local masking and downsampling. Upon visualizing the degraded image across different noise levels, we performed experiments on three different variances $\sigma = 0.05, \sigma = 0.1, \sigma = 0.2$. It can be observed that the downstream performance is the best while $\sigma = 0.1$. The results can be interpreted in an intuitive manner as it's known that a higher amount of noise hyper-parameter will degrade the image significantly and thus makes it more difficult for the model to learn any useful low-level representations. While a low amount of noise does not help in learning good representations as the image hardly goes through any kind of degradation and is likely to be directly copied by the network to get the reconstruction.

Table 8: Impact of noise level. Dice scores are from experiments done on first fold of BTCV dataset.

| Noise Level | Dice ($\uparrow$) |
|---|---|
| $\sigma = 0.05$ | 84.56 |
| $\sigma = 0.1$ | 85.90 |
| $\sigma = 0.2$ | 85.43 |

### G.2. Impact of downsampling factor

In Table 9, we study the impact of downsampling ratio during pre-training on the fine-tuning performance. Note that in this experiment, we do not include other perturbations of local masking and adding noise. The parameter $\epsilon$ is the factor of downsampling operation done. Upon visualizing the degraded image across $\epsilon$, we performed experiments on three different variances $\epsilon = 2, \epsilon = 4, \epsilon = 8$. It can be observed that the best performance is when $\epsilon = 4$. The reason here is also similar as that of noise level as a higher perturbation makes it difficult to learn useful representation while lower perturbation does not distort the image much hence the network just learns to copy information from input to reconstruction.

Table 9: Impact of downsampling ratio. Dice scores are from experiments done on first fold of BTCV dataset.

| Downsampling ratio | Dice ($\uparrow$) |
|---|---|
| $\epsilon = 2$ | 85.3 |
| $\epsilon = 4$ | 85.64 |
| $\epsilon = 8$ | 85.18 |

### G.3. Impact of local masking ratio

In Table 10, we study the impact of masking ratio of local masking during pre-training. The parameter $r$ is the factor of downsampling operation done. Note that in this experiment, we do not include other perturbations of downsampling and adding noise. We observe a good performance when masking ratio is 0.4 while the performance decreases as we increase it.

Table 10: Impact of masking ratio of local masking. Dice scores are from experiments done on first fold of BTCV dataset.

| Masking Ratio | Dice ($\uparrow$) |
|:---:|:---:|
| R = 0.4 | 85.55 |
| R = 0.6 | 84.13 |
| R = 0.8 | 83.71 |

