# OpenReview forum: "Disruptive Autoencoders: Leveraging Low-level features for 3D Medical Image Pre-training"
_MIDL.io/2024/Conference — MIDL 2024 Oral_

### Official Review · Reviewer_wG9x · 2024-02-26

**Confidence:** 4
**Preliminary Rating:** 5
**Recommendation:** Oral

**Summary:**

This work presents a pre-training framework for 3D radiology images. They propose a local masking strategy, and combine it with low-level perturbations to further enable low-level representation learning. The authors curate a large-scale dataset of 10,000 3D medical radiology images (MRI and CT) for pre-training. Multiple downstream tasks are tested achieving state-of-the-art.

**Strengths:**

- Paper is well written and easy to follow.
- Code is shared and documented in a GitHub repository.
- Figures 1-3 provide a comprehensive view of the proposed method.
- Proposed method is compared against other methods and outperforms them. An ablation study showing the effectiveness of the different disruption strategies is provided, as well as the modal contrastive learning part.

**Weaknesses:**

- Ablation study. Could you provide clarification regarding the dataset to which the results in Table 2 correspond?
- The main paper lacks discussion of limitations of the proposed method and ideas for future work.

**Detailed Comments:**

Minor improvements:
- Figure 5 and 6 seems to be Tables, the use of Figure / Table seems ambiguous in some cases
- The caption of Figure 5 could include the name of the dataset used for evaluation for clarity
- There is no reference to Table 1.

Suggestions:
- Sharing the curated collected dataset (MRI + CT) would be of great value for the medical imaging research community, and strengthened the transparency of the results.

**Justification Of The Preliminary Rating:**

The authors review the relevant related work, they motivate and propose a novel method. The manuscript is well structured and well written. Their work is extensively validated. Some minor improvements could be considered.

**Questions To Address In The Rebuttal:**

I believe this work has been extensively validated and the manuscript is well structured and well written. I would suggest a discussion about the limitations of the proposed method, future work and availability of the curated dataset.

**Special Issue:**

Yes

---

> ### Author Response · Authors · 2024-03-18
> **Official Reply to Reviewer wG9x**
>
> ***1) Ablation Study***
>
> The experiments of ablation study were conducted on a five-fold cross validation of BTCV dataset. We have made sure to clearly state this in the revised version of the paper.
>
> ***2) Limitations and future work***
>
> We would like to point the reviewer to the Appendix E for limitations and future works. We had included this in our main submission but have now added more points to it.
>
> There is  scope for improvement in picking the best combination of $\sigma, \epsilon, r$ for the pre-trained weights. A grid search on these parameters would be time-consuming but would lead to better pre-trained weights. Moreover, it can be noted that it takes a lot of time and computation to conduct pre-training experiments with large-scale datasets on dense medical volumes. There is also scope for improvement in increasing the size of the pre-training dataset to obtain more generalizable weights. For future work, we plan on expanding to more modalities like US, XRs, etc.
>
> ***3) Availability of Dataset***
>
> The dataset names and links will be provided along with our code-repository. Due to some policies with TCIA datasets that prevent use from re-sharing it with a new link, we would not be able to share our curated dataset as a single link. However, we will make sure have all the separate links and a readme to help future researchers easily use the pre-training data we used.
>
> ***4) Figure/Table and references to them***
>
> We thank the reviewer for this comment and in the revised edition, we have used only Table captions for Tables and Figure captions for Figure. We have also re-structured the paper and cross-links to improve readability.

---

> ### Comment · Reviewer_wG9x · 2024-03-26
>
> Thanks to the authors for the revised manuscript. Unfortunately, the changes or additions haven't been highlighted or colored, so this makes it difficult to review them. I am satisfied with their answers to my minor comments in OpenReview and I don't have any further comments.

---

### Official Review · Reviewer_5JqC · 2024-03-01

**Confidence:** 5
**Preliminary Rating:** 1

**Summary:**

This paper introduces a self-supervised pretraining approach for 3D medical images. The proposed method is primarily based on masked image modeling (MAE), and uses different image augmentations, including channel-wise masking, downsampling, and noise injection to perturb the images. The proposed method also utilizes a contrastive learning loss. Evaluation of the proposed method has been conducted on BTCV and FeTA datasets.

**Strengths:**

The proposed method achieves top results in the BTVC leaderboard.

The authors conducted ablation studies on three augmentations used in the paper.

The authors conducted ablation study on different loss terms.

**Weaknesses:**

1. The novelty of the paper is limited as the proposed method largely overlaps with existing reconstruction techniques, such as [1,6], and merely introduces minor alterations concerning image augmentation. Additionally, the integration of contrastive learning with reconstruction has been previously explored in existing self-supervised learning methods, such as  [4] and Swin-UNETR (Tang et al.).

2. The proposed method lacks clear motivation, and the rationale behind its design choices is not adequately justified or supported by results.

   * The paper advocates for local masking to enhance the reconstruction capability of MAE, even though achieving optimal reconstruction is not the primary objective of MAE or other reconstruction-based SSL methods. Furthermore, there are no experimental results provided to demonstrate the efficacy of local masking standalone over conventional masking in MAE.
   *  The rationale behind the cross-modal contrastive loss is unclear. Despite being labeled as cross-modal, it operates across augmented versions of the same images rather than different modalities, making it similar to conventional contrastive learning. Moreover, there is a lack of experimental results demonstrating the superiority of this approach over contrastive loss.

3. The literature review is outdated and overlooks the most relevant and recent works, including [1,2,3,4,5]. I recommend the authors conduct a thorough literature review to provide a comprehensive and up-to-date review of  the latest advancements in self-supervised learning in medical imaging, which is necessary for identifying the original contributions of the paper and its advantages over existing methods, and selection of more appropriate baselines.

4. The experimental results fail to sufficiently demonstrate the effectiveness of the proposed method and support the claim of state-of-the-art performance.

   * The authors state that the proposed method outperform Swin-UNETR (Tang et al., 2022), which also utilizes SSL pre-training. However, it's important to note that Swin-UNETR leverages 5K pretraining data, whereas the proposed method benefits from more than 10K data. Therefore, this comparison may not be entirely fair and does not convincingly demonstrate the superiority of the proposed pretraining method over the  Swin-UNETR’s pretraing schema.
   * Continuing from the previous point, there is a lack of comparison with Swin-UNETR (Tang et al., 2022) in the FeTA dataset. It is necessary to compare the proposed method with Swin-UNETR in the same pretraining setting in terms of pretraining data in both target tasks.
   * The authors consider MAE as the main baseline; however, it does not appear to be a competitive baseline. For instance, as depicted in Figure 6, the performance of MAE is even inferior to training from scratch.
   * Continuing from previous point, the performance boost over MAE in FeTA dataset seems to be marginal, and there is no statistical analysis to demonstrate the significance of the results.
   *  The proposed method has not been compared with state-of-the-art self-supervised methods specifically designed for medical imaging, such as [1,2,3,4,5]. Instead, it has been compared with basic SSL methods such as rotation, and just MAE in FeTA dataset.

5. The results in Table 4 in the Appendix raises concerns regarding the effectiveness of the proposed method in learning features suitable for downstream applications. The reported CKA scores are notably low, indicating potential limitations in learning reusable features. Additionally, the proposed DAE provides lower reusable features across all layers, with the exception of layer 1. Consequently, this result does not convincingly establish the superiority of the proposed DAE over MAE in capturing useful features for medical images.

6. There is no reference in the text to Figure 6, and its results are not discussed in the paper.

7. The performance of the proposed DAE varies across Tables 1, 2, and 3, which is confusing. There is a lack of explanation regarding the cause of this discrepancy.


[1] Masked Image Modeling Advances 3D Medical Image Analysis, 2023

[2] Towards Foundation Models Learned from Anatomy in Medical Imaging via Self-Supervision, 2023

[3] Joint Self-Supervised Image-Volume Representation Learning with Intra-inter Contrastive Clustering, 2023

[4] DiRA: Discriminative, Restorative, and Adversarial Learning for Self-supervised Medical Image Analysis, 2022

[5] LVM-Med: Learning Large-Scale Self-Supervised Vision Models for Medical Imaging via Second-order Graph Matching, 2023

[6] Models Genesis, 2021.

**Detailed Comments:**

1. The impact of downsampling augmentation seems to be marginal. How the model would perform if remove it and just use the other two augmentations?

2. The paper's organization needs improvement to make it more concise and clear.
  * The related work section has been placed in the Appendix, although there are unnecessary details in other sections which can be condensed in order to incorporate related works section into the main paper, thereby ensuring its completeness.
  * The authors refer to Table 4 and Table 5, while there are no such tables in the paper.
  * The cross links in the paper point to nowhere when click on them, e.g "2" in caption of Figure 5, Table 5, etc.
  * Some tables has been indexed as figure, while others have been indexed as table.
 * The index of the tables does not follow their reference order in the paper.

**Justification Of The Preliminary Rating:**

The paper’s novelty is limited and its contributions may not be significant. The proposed method lacks clear motivation, and the rationale behind its design choices is not adequately justified or supported by results. Additionally, the experimental setups have flaws and fail to demonstrate the efficacy and generalizability of the proposed method. Furthermore, the related works are outdated and overlook the most relevant and recent studies, which has also led to the selection of less competitive baselines.

**Questions To Address In The Rebuttal:**

Please refer to the weaknesses section.

---

> ### Author Response · Authors · 2024-03-18
> **Official Reply to Reviewer 5JqC**
>
> ***1) Novelty***
>
> The reviewer mentions that our work “merely introduces minor alterations” which we respectfully disagree with. Our method is novel in the following ways:
>
> **a) Generalized starting weights for multiple modalities:** Unlike any previous medical pre-training works, we provide starting weights that could be used for downstream medical tasks of different modalities proved by the fact we use the same pre-trained weight for all our results.
>
> **b) Local Masking:** We would like to highlight that local masking is not similar to MAE. MAE masks out around 75\%  patches, which will inevitably remove all information across all the regions of the image, and that information needs to be recovered from far away patches and forcing it to learn more global information. Our local masking keeps part of the information for every patch thus forcing the model to see more locally. This forces the model to learn fine-grained information. This is a fundamental change and we showed that it is useful for medical imaging through empirical results as well as sharper reconstructions than MAE as seen in Fig 2. We further direct the reviewer to read Appendix B.
>
> **c) Specialized  3D medical image pre-training:** We develop a new pre-training pipeline considering the reconstruction of fine-grained details of medical images by utilizing low-level perturbations and local masking. Noise and low resolution are commonly found issues in realistic clinical medical acquisition pipelines and we make use of it to develop an effective medical image pre-training pipeline and this has not been done in previous medical pre-training works.
>
>
> ***2) Motivation***
>
> We agree with the reviewer that achieving optimal reconstruction is not the primary objective of MAE or other reconstruction-based SSL methods. However, please note that most reconstructions not really capturing the fine-grained details could potentially cause the encoder to not learn useful information.  In our work, we observed that if the reconstruction quality is good, the downstream performance is also good given the input image is perturbed enough. There is a sweet spot between the perturbation level and reconstruction where the downstream performance is optimal. We find this from experiments reported in Table 8,9,10 of appendix.
>
> ***3) DAE Results across Tables 1,2, and 3 (revised: 4,5, and 6)***
>
> To maintain consistency, now we have reported the results on Table 4 and 5 on the five-fold BTCV cross validation dataset. For Table 6, we just did the experiment on a single fold of BTCV validation due to time constraints.
>
> ***4) Local Masking vs MAE***
>
> The reviewer also mentions that “there are no experimental results provided to demonstrate the efficacy of local masking standalone over conventional masking in MAE”. For this, we now add MAE comparison to Table 5. It can be seen that Local Masking is directly better than MAE.
>
>
> ***5) Literature Review***
>
> We thank the reviewer for pointing out the most recent works. We have added them to the revised version. We would like to point out that [1] was cited in our paper and is actually compared to as well in our experiments (as MAE).  We also cited [6] in the paper.
>
> ***6) MAE as a baseline***
>
> MAE is in fact one of the strongest baselines for pre-training in computer vision. Also, it can be argued masking-based techniques are more efficient than [2,3,4,5] as it is faster (due to masking x% of tokens) leading to quadratically reducing the computation while training with masked tokens. That being said, one main aspect of our paper is to show the MAE is not an optimal pre-training strategy for medical image pre-training. We did a fair job in training MAE-based pre-training and we do report our findings that MAE is sometimes worse than training from scratch.
>
> ***7) Statistical Significance of MAE boost in FeTA dataset***
>
> While our improvement is 2.6%, 0.9%, and 0.4% in terms of dice accuracy across different distributions of data, we conduct a paired t-test between our FETA test Dice scores and MAE's results. We obtain a two-tailed p-value of 0.0258 which shows our improvement is statistically significant (p < 0.05).

---

> > ### Author Response · Authors · 2024-03-18
> > **Continuation of Official Reply to Reviewer 5JqC**
> >
> > ***8) On CKA values being less in deeper layers***
> >
> > Contrary to the reviewer’s opinion, we believe that this observation on CKA values being less in deeper layers actually helps strengthen our claim that i) low level features are more important for  medical image pre-training ii) DAE is better than MAE at extracting low-level details.
> >
> > To break this down, the early layers have a higher CKA meaning more low-level feature representations were retained from the pre-trained weights to the fine-tuned model. As high-level features undergo a relatively heavier change for both MAE and DAE compared to low-level features after fine-tuning, therefore it is important to focus on learning stronger low-level features during pre-training. Second, we can observe that the CKA values of DAE are larger than MAE, especially in the early layers. This proves that more low-level features are retained for DAE, showing that DAE is better at learning low-level feature representations.
> >
> >
> >
> > ***9) Contrastive loss***
> >
> > The reviewer also misunderstands the contrastive loss proposed in our work and compares them with [4] and Swin-UNETR (Tang et al.). While the concept of contrastive loss is not new, our loss is devised to distinguish modalities in the latent space to make the pre-trained encoder modality aware.
> >
> >
> > ***10) Organization of the paper and cross-links***
> >
> > In the revised edition, we have made the structure of the paper better and made sure all the Figures and Tables are cross linked in the text.

---

> > > ### Author Response · Authors · 2024-03-26
> > >
> > > We believe that we have thoroughly addressed the reviewer's concerns and kindly request the reviewer to let us of any additional issues following our response. We thank the reviewer for their time.

---

### Official Review · Reviewer_8m97 · 2024-03-06

**Confidence:** 3
**Preliminary Rating:** 4
**Final Rating:** 5

**Summary:**

This paper proposes a generic framework to pretrain neural networks for 3D MRI and 3D CT, which consists in a transformer-based auto-encoder, coined Disruptive Autoencoder (DAE).
The tokens are disrupted with different perturbations, including a proposed Local Masking.
The usefulness of the DAE is assessed with a segmentation task, and compared to other methods.

**Strengths:**

The paper proposes an interesting way to pre-train 3D MRI and CT images neural network.
The results on the segmentation task are good, and the ablation study shows that the proposed Local Masking is useful.

**Weaknesses:**

The paper is not well structured, and a bit hard to follow.
Some figures are not used (5 and 6), only the first line of figure 2 is used.
The order of the figures is a bit odd.
The Local Masking is properly explained in section 2. Before that two references to Fig 1 and 3 are made to explain it, but the figures are not clear enough by themselves.

**Detailed Comments:**

Acronyms MAE and SimMIM are defined only in the appendix.

Fig. 1 and the corresponding text is unclear on the perturbations:
Fig. 1 suggests that you add the LR patch to the HR noisy patch. The text says that you “combine” it. What do you do exactly?

Figure 2 is interesting, but not really used in the paper.

In 2, N is the number of tokens. In the training, is your data points N the number of tokens?

There is no z in Eq (4).

In 3.2, 20K is written instead of 20k.

In the results, it is not clear if it uses only the BTCV fine-tuning or the FeTA too.

Figure 5 and figure 6 are not cited in the text.

Acronym in 3.3 SSL is not defined.

**Justification Of Final Rating:**

The structure of the paper has clearly improved, and is now much more easy to follow.
The method is interesting, the results are clearly presented, and show some improvement compared to previous methods.

**Justification Of The Preliminary Rating:**

The proposed Local Masking is interesting, as shown in the comparison of reconstructions in Figure 2.
Indeed, the results are improved compared to the Masked Autoencoders.
However, the paper is not well structured, and thus hard to follow, and the segmentation experiment is not clear.

**Questions To Address In The Rebuttal:**

Please improve the structure of the paper to make it easier to follow.

---

> ### Author Response · Authors · 2024-03-18
> **Official Reply to Reviewer 8m97**
>
> ***1) Combining LR Patch to HR noisy Patch***
>
> We apologize to the reviewer for the confusion. By “combine” we still meant “add”. Please note that the LR Patch is added to the noisy HR patch. To avoid any confusion, we have corrected "combine" to "add" in the revised edition.
>
> ***2) Figure 2 and the point it makes***
>
> In Figure 2, we compare the reconstructions obtained from various pre-training techniques like masking, denoising, super-resolution, local masking, and DAE. This figure shows how masked image modeling produces a  coarse reconstruction for radiology without local context while the proposed disruptions in this work obtain sharper reconstructions recovering meaningful fine details.
>
> ***3) Clarification of N being data points and the number of tokens***
>
> We apologize to the reviewer for the confusion. In 2, N corresponds to the number of tokens.  In training, we have now changed it to K to correspond to the number of data points to maintain consistency.
>
> ***4) About z and Equation 4***
>
> We have removed the typo and have corrected it: zlabel corresponds to the labels.
>
> ***5) About Results***
>
> We show results on BTCV (Table 1) as well as FeTA datasets (Table 2).
>
> ***6) Structure and order of the figures***
>
> We apologize to the reviewer for the presentation's mishaps and have made necessary changes in the revised edition to improve readability, structure and the order of figures match the order we talk about them in the text. We have also added details to Figures/Tables for which explanation was missing in the text.
>
> ***7) Other Typos***
>
> We have also corrected all the other typos mentioned by the reviewer in the revised edition.

---

> > ### Comment · Reviewer_8m97 · 2024-03-26
> >
> > Thank you for your answers.

---

### Author Response · Authors · 2024-03-18
**General Comment to the AC and Reviewers**

We would like to sincerely thank all the reviewers for their clear, detailed, and genuine comments on our paper. In the comments below, we have addressed all the issued the reviewers had. We also made necessary changes in the revised edition of the paper. We believe the inputs from the reviewers has made our revised version better, and we again thank the reviewers for that.

---

### Meta-Review · Area_Chair_N3JA · 2024-04-04

**Recommendation:** Accept (Poster)
**Confidence:** 4

**Metareview:**

The work has wide applicability to medical imaging due to its ability to deal with multiple modalities. A large dataset is used and results in significant improvements. The authors engaged in the rebuttal with convincing answers.

---

### Decision · Program_Chairs · 2024-04-06

Accept (Oral)